# A widely employed germ cell marker is an ancient disordered protein with reproductive functions in diverse eukaryotes

Michelle A Carmell[1]*, Gregoriy A Dokshin[2], Helen Skaletsky[1,3], Yueh-Chiang Hu[1†], Josien C van Wolfswinkel[1‡], Kyomi J Igarashi[1], Daniel W Bellott[1], Michael Nefedov[4§], Peter W Reddien[1,3,5], George C Enders[6], Vladimir N Uversky[7], Craig C Mello[2,3], David C Page[1,3,5]*

[1]Whitehead Institute, Cambridge, United States; [2]RNA Therapeutics Institute, University of Massachusetts Medical School, Worcester, United States; [3]Howard Hughes Medical Institute, Chevy Chase, United States; [4]BACPAC Resources, Children's Hospital Oakland, Oakland, United States; [5]Department of Biology, Massachusetts Institute of Technology, Cambridge, United States; [6]Department of Anatomy and Cell Biology, University of Kansas Medical Center, Kansas City, United States; [7]Department of Molecular Medicine, Morsani College of Medicine, University of South Florida, Tampa, United States

*For correspondence: carmell@ wi.mit.edu (MAC); dcpage@wi.mit. edu (DCP)

Present address: †Cincinnati Children's Hospital Medical Center, Division of Developmental Biology, Cincinnati, United States; ‡Department of Molecular, Cellular and Developmental Biology, Yale University, New Haven, United States; §School of Chemistry and Molecular Biosciences, University of Queensland, Brisbane, Australia

Competing interests: The authors declare that no competing interests exist.

**Abstract** The advent of sexual reproduction and the evolution of a dedicated germline in multicellular organisms are critical landmarks in eukaryotic evolution. We report an ancient family of GCNA (germ cell nuclear antigen) proteins that arose in the earliest eukaryotes, and feature a rapidly evolving intrinsically disordered region (IDR). Phylogenetic analysis reveals that GCNA proteins emerged before the major eukaryotic lineages diverged; GCNA predates the origin of a dedicated germline by a billion years. *Gcna* gene expression is enriched in reproductive cells across eukarya – either just prior to or during meiosis in single-celled eukaryotes, and in stem cells and germ cells of diverse multicellular animals. Studies of *Gcna*-mutant *C. elegans* and mice indicate that GCNA has functioned in reproduction for at least 600 million years. Homology to IDR-containing proteins implicated in DNA damage repair suggests that GCNA proteins may protect the genomic integrity of cells carrying a heritable genome.

## Introduction

Sexual reproduction in eukaryotes is accomplished through meiosis, a specialized cell cycle that generates genetically diverse derivatives. Meiosis is believed to have evolved once, in the common ancestor of extant eukaryotes, about two billion years ago (*Ramesh et al., 2005*; *Villeneuve and Hillers, 2001*). At the same time, special protection for heritable genomes evolved in the form of an RNAi pathway that likely functioned in defense against transposons and viruses (*Cerutti and Casas-Mollano, 2006*; *Shabalina and Koonin, 2008*). Sexual reproduction via meiosis and utilization of specialized genome defense mechanisms are characteristics of many unicellular eukaryotes, in which every cell contributes to the next generation. These properties became germline specific when the distinction between germ cells and somatic cells was made in multicellular organisms, with the germline being tasked with guarding the heritable genome. Accordingly, certain critical genes encoding proteins associated with the sexual cycle, as well as those required for integrity of the heritable

**eLife digest** Sex allows two individuals of a species to combine their genetic information and generate new offspring. As such, and unlike most of the cells that make up an organism, reproductive cells carry DNA that will be passed to the next generation. To prepare for sexual reproduction, reproductive cells undergo a process whereby they give up half of their DNA. Moreover, in many species, reproductive cells undergo dramatic changes in size and shape to become specialized as eggs or sperm.

For two decades, research scientists have made use of an antibody that specifically targets and labels reproductive cells in mice and rats, long before they become recognizable as sperm or eggs. However, nothing was known about the protein that this antibody recognizes.

Now, Carmell et al. have found that the antibody recognizes a protein called GCNA. This protein was previously thought to be limited to mice and rats. However, GCNA is in fact a member of a far-flung family of proteins that had gone unnoticed by researchers due to their unusual composition and large unstructured regions. Carmell et al. show that this protein family extends beyond rodents and even mammals, and actually exists in all branches of eukaryotic life, including plants and single-celled organisms. In most species examined, the genes encoding these proteins are most active in reproductive cells.

The most important next step is to figure out precisely what GCNA proteins are doing in reproductive cells. This will also hopefully explain why these proteins appear to have been conserved for more than 2 billion years, throughout the entire history of eukaryotes.

genome, share a deep, common evolutionary origin (*Uanschou et al., 2007*; *Kumar et al., 2010; Cerutti and Casas-Mollano, 2006*). In metazoans, this reproductive assemblage expanded to encompass proteins not directly involved in meiosis but nonetheless expressed exclusively in germ cells; these include the RNA binding proteins DAZL/BOULE, VASA, and NANOS (*Ewen-Campen et al., 2010*; *Juliano et al., 2010*).

Mouse is the most widely used model to study the germline in mammals. Across 22 years of research, and >425 publications in mouse reproductive biology, investigators have employed antibodies to two antigens – GCNA1 and TRA98 – to distinguish germ cells from somatic cells. Nonetheless, the identity of the antigens themselves has remained unknown. The striking qualities of these markers, together with their importance to the research community, led us to attempt to discover the underlying antigens. Here, we identify the antigen recognized by both antibodies as a single, unannotated protein in the mouse. We name this protein GCNA and describe its ancient and ubiquitous association with sexual reproduction.

## Results

### GCNA1 and TRA98 antibodies recognize the same previously unannotated protein

The mouse germline is first distinguished from the surrounding soma as a population of primordial germ cells that migrates toward the developing genital ridge. Upon arrival, germ cells undergo a critical transition and commit to sexual differentiation, eventually giving rise to oocytes or spermatozoa. The handful of proteins that are expressed concomitant with entry into the genital ridge include the classical germ cell factors DAZL and DDX4 (Mouse Vasa Homolog; MVH) and the antigens recognized by GCNA1 and TRA98 antibodies (*Figure 1A*) (*Hu et al., 2015*; *Enders and May, 1994*; *Tanaka et al., 2000*).

The GCNA1 and TRA98 monoclonal antibodies, generated independently from rats immunized with cell lysates from adult mouse testis, are robust markers of mouse germ cell nuclei and show no reactivity to somatic cells (*Enders and May, 1994*; *Tanaka et al., 2000*). To clone the GCNA1 antigen, we carried out immunoprecipitation from an adult mouse testis lysate, followed by mass spectrometry. We detected 26 unique peptides representing 51% coverage of an unannotated protein specifically in the immunoprecipitate, enabling us to confidently identify it as GCNA (*Figure 1B*,

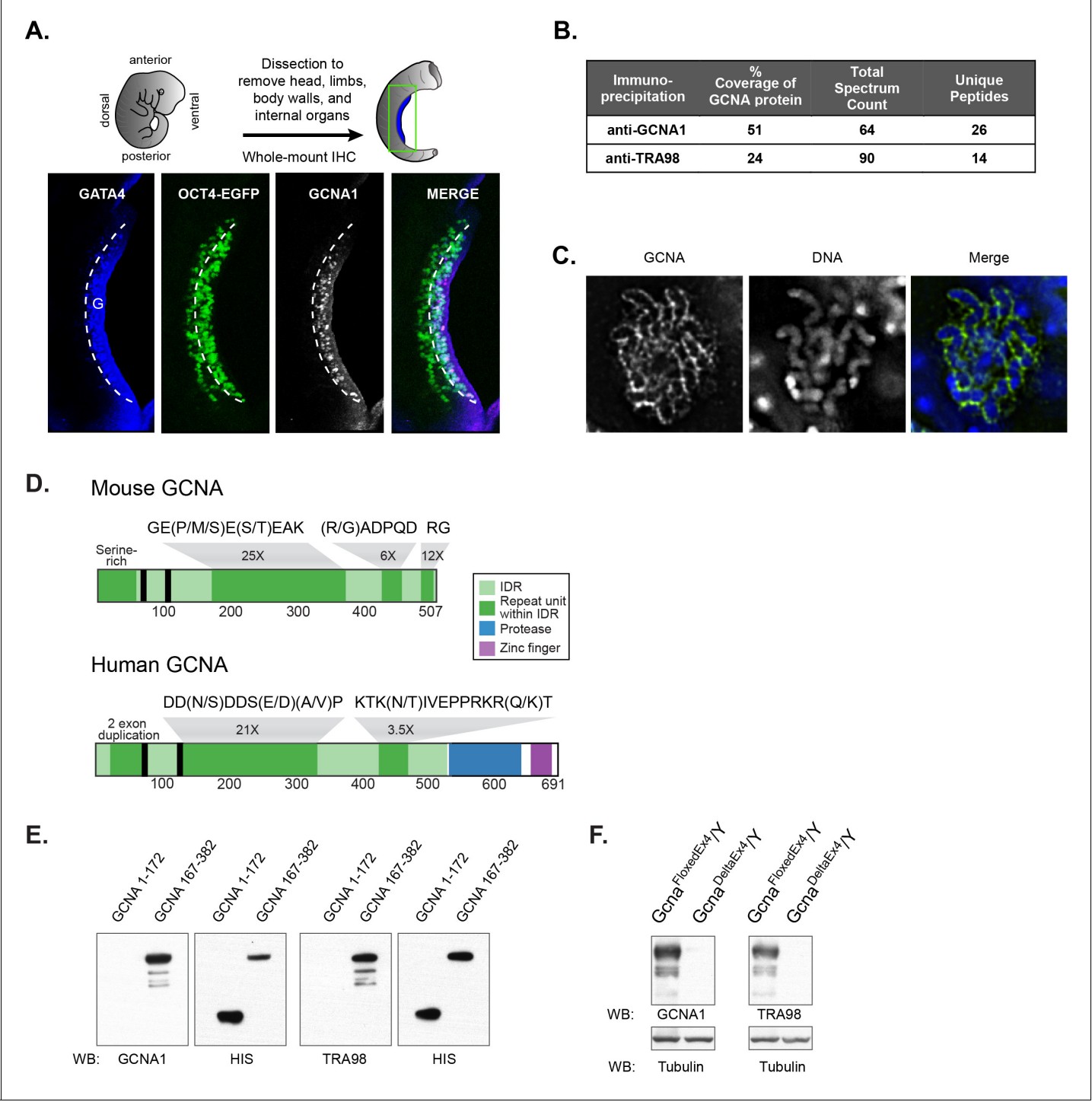

**Figure 1.** Identification of the antigen recognized by GCNA1 and TRA98 antibodies. (A) Schematic illustration depicting isolation of a mouse embryonic day 11.5 gonad. GCNA (white, pink in merge) is specifically expressed by germ cells (green) only after they enter the somatic gonad (G), which is marked by its expression of GATA4 (blue). Dashed line indicates boundary of gonad. Oct4-EGFP transgene labels germ cells. (B) Mass spectrometry indicates that GCNA1 and TRA98 antibodies immunoprecipitate the same protein. (C) GCNA coats condensed chromosomes during meiotic prophase. One plane through a single nucleus of a male spermatocyte in the zygotene phase of Meiosis I is shown. (D) Mouse GCNA encodes an acidic and repetitive protein that is predicted to be disordered. Intrinsically disordered regions (IDRs) are depicted in green; repeating units of the indicated sequences are shown in dark green and non-repetitive disordered sequences are light green. Human GCNA shares a region with the same acidic and repetitive character (green) and contains additional, ordered domains – a protease domain (blue), and a C2C2 zinc finger (purple). SUMO-interacting motifs (SIMs) are indicated by black bars. (E) Bacterially expressed recombinant HIS-tagged mouse GCNA is recognized by both GCNA1

*Figure 1 continued on next page*

*Figure 1 continued*

and TRA98 antibodies. The antigen is located within a protein fragment containing the murine-specific GE(P/M/S)E(S/T)EAK repeat. (**F**) Western blot of mouse XY embryonic stem cell lysate with a disruption in the identified locus demonstrates depletion of both GCNA1 and TRA98 antigens.

The following source data and figure supplements are available for figure 1:

**Source data 1.** Mass spectrometry data.
**Figure supplement 1.** Characterization of mouse *Gcna* transcript and protein.
**Figure supplement 2.** Generation of *Gcna*-targeted ES cells and mice.

*Figure 1—source data 1*). Mouse GCNA contains four distinct repeat classes that comprise the majority of the protein, and its theoretical isoelectric point of 4.17 makes it more acidic than 98.9% of all mouse proteins (*Figure 1D*, *Figure 1—figure supplement 1*) (*Bjellqvist et al., 1993*).

The developmental timing and cell type specificity of labeling with GCNA1 resembles that of TRA98, a second antibody with an unknown antigen (*Tanaka et al., 2000*; *Inoue et al., 2011*). The subcellular localization of GCNA1 and TRA98 also show striking similarities; we find that GCNA forms a distinctive coating around condensed chromosomes in meiotic prophase (*Figure 1C*), and TRA98 has been noted to have a similar reticular or netlike localization in the nucleus (*Inoue et al., 2011*). Due to these parallels, we hypothesized that the TRA98 antibody recognized the same antigen as GCNA1. Indeed, immunoprecipitation using TRA98 yielded 24% coverage of the GCNA protein (*Figure 1B*, *Figure 1—source data 1*). By expressing portions of mouse GCNA in bacteria, we determined that both antibodies recognize a fragment containing a murine-specific 8-amino-acid tandem GE(P/M/S)E(S/T)EAK repeat that occurs 25 times in the protein (*Figure 1D,E*). Additionally, we disrupted the gene encoding GCNA in mouse embryonic stem (ES) cells (*Figure 1—figure supplement 2*) and found that antigens recognized by both antibodies were depleted, confirming that GCNA1 and TRA98 antibodies recognize the same protein (*Figure 1F*).

## Mouse GCNA is predicted to be entirely disordered

The repetitive structure and biased amino acid composition of mouse GCNA is characteristic of intrinsically disordered protein regions (IDRs). IDRs display conformational flexibility and have no single, well-defined equilibrium structure, and yet carry out numerous biological activities (*van der Lee et al., 2014*). IDRs have high absolute net charge due to enrichment for disorder-promoting (charged and polar) amino acids, and low net hydrophobicity due to depletion of hydrophobic and order-promoting amino acids, features that make it possible to predict disordered regions from primary amino acid sequence alone (*Uversky et al., 2000*; *He et al., 2009*). Based on its extreme negative charge and atypical amino acid composition, mouse GCNA is predicted to be entirely disordered (*Figure 1—figure supplement 1*, *Figure 2A*).

## GCNA orthologs containing an IDR and a unique combination of conserved structured domains are present in every eukaryotic superkingdom

Alignment-based searches using the amino acid sequence of mouse GCNA as bait failed to detect significant similarity to any protein in any species, suggesting that GCNA might be unique to mice. Alternatively, GCNA could be rapidly evolving, as is common for disordered proteins due to their low level of structural constraint (*Brown et al., 2010*; *Huntley and Golding, 2000*; *Tompa, 2003*). To distinguish between these scenarios, we sought to identify GCNA orthologs in other species.

Using a combination of methods that accommodate rapid evolution, we were able to identify GCNA orthologs in a broad swath of organisms from human to the most primitive single-celled eukaryotes. Because rapid evolution often renders IDRs un-alignable even among closely related species (*Brown et al., 2010*), primary sequence cannot be used to establish orthology. Therefore, starting with mouse GCNA, which is encoded by a gene on the X chromosome, we identified orthologs in human and other vertebrates by synteny (*Figure 3*). We discovered that, whereas mouse GCNA is predicted to be entirely disordered, many GCNA orthologs, including the human ortholog,

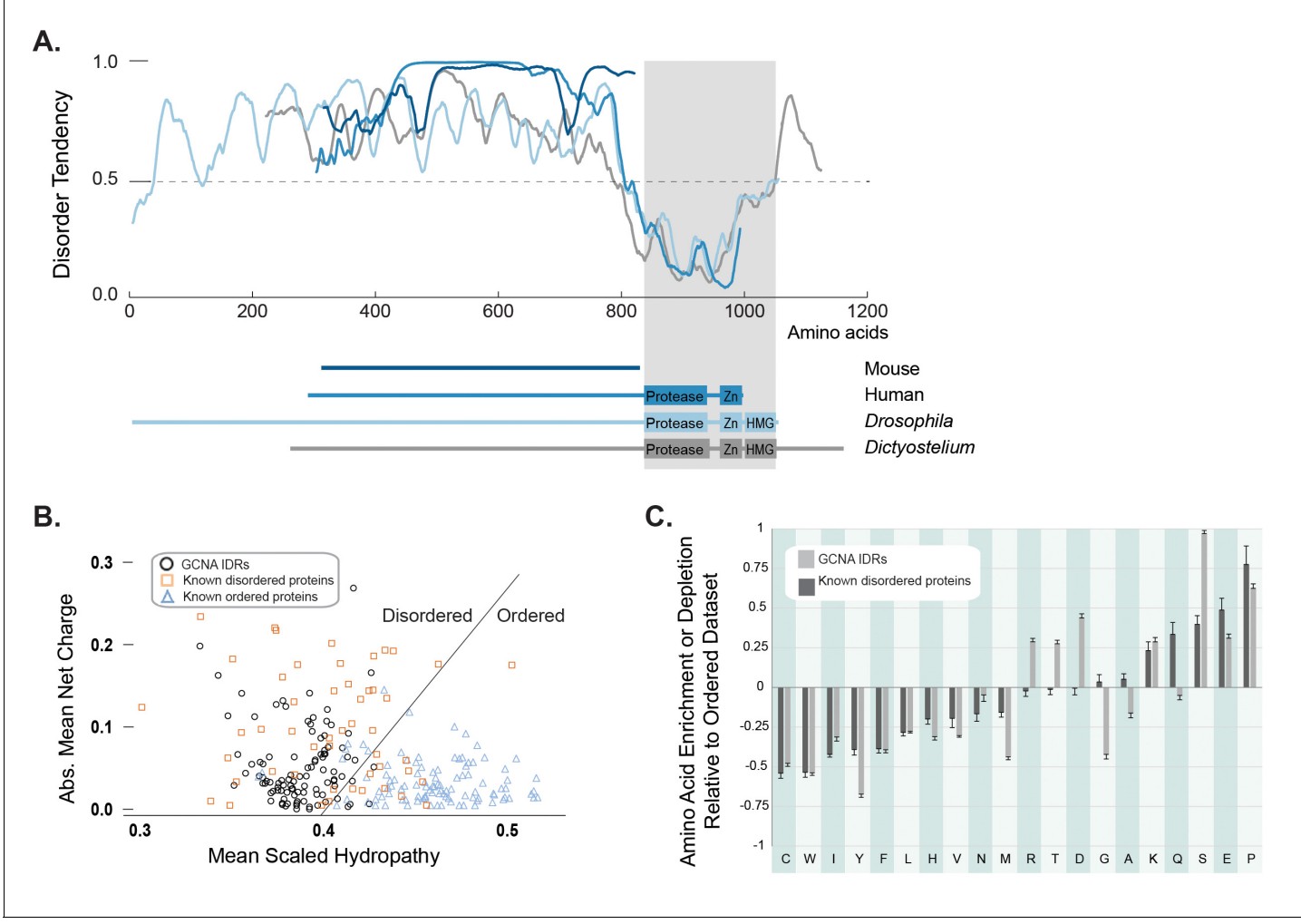

**Figure 2.** GCNA proteins across eukarya are predicted to have large intrinsically disordered regions. (**A**) Disorder tendency of GCNA proteins from mouse (navy), human (medium blue), *Drosophila melanogaster* (light blue), and *Dictyostelium discoideum* (gray). Residues above the dotted line are predicted to be disordered. Folded domains are indicated by gray rectangle. (**B**) Charge-hydropathy analysis (mean scaled hydropathy, <H>, against absolute mean net charge, <R>) predicts GCNA IDRs to be disordered by comparison to a set of known disordered proteins (orange squares) and ordered proteins (blue triangles). IDRs of 114 GCNA proteins across eukarya (black circles) are plotted. The boundary between unfolded and folded space is empirically defined by the equation <H>b = (<R> + 1.151)/2.785 (*Uversky et al., 2000*). (**C**) Amino acid composition of GCNA IDRs. Enrichment or depletion is expressed as $(C_x - C_{order})/C_{order}$, representing the normalized excess of a given residue's content in a query dataset ($C_x$) relative to the corresponding value in the dataset of ordered proteins ($C_{order}$). Error bars represent fractional differences of the standard deviations of observed relative frequencies of bootstrapped samples of the datasets.

The following source data is available for figure 2:

**Source data 1.** Charge/hydropathy analysis.

have well-conserved structured domains in addition to an IDR (*Figure 1D*, *2A*). Specifically, we deduced that three domains had been lost more than 25 million years ago in the rodent lineage (*Figure 3—figure supplement 1*). Non-transcribed pseudo-exons that previously encoded these domains are found in the mouse genome, allowing us to confidently place mouse GCNA into this larger family (*Figure 3—figure supplement 1*).

To explore the phylogenetic reach of the GCNA family, we used the conserved regions in a core set of vertebrate GCNA proteins to identify more distant orthologs by sequence similarity. We created statistical profiles (hidden Markov models or HMMs) of the sequence of each conserved region (*Figure 4—figure supplement 1*). We found that each region has a unique signature found only in

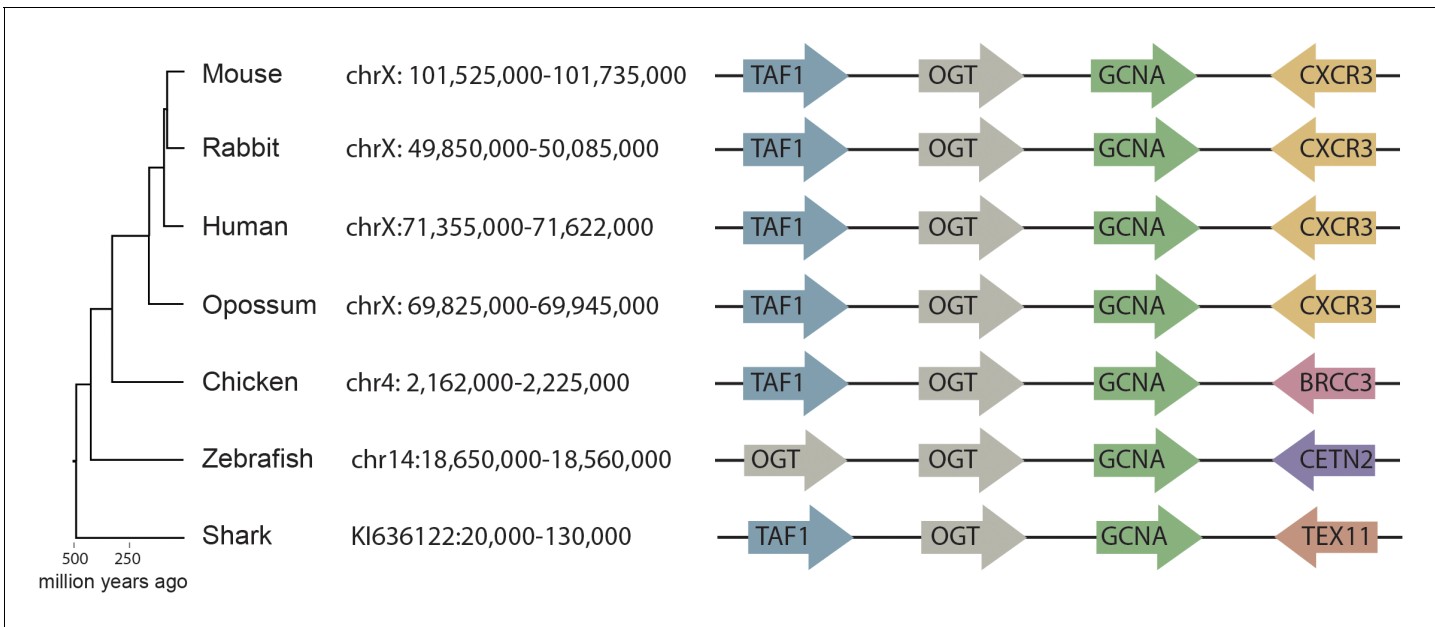

**Figure 3.** Synteny mapping of *Gcna* orthologs across vertebrates. Comparison of the genomic region containing *Gcna* in mouse (GRCm38/mm10) with syntenic regions in human (GRCh38/hg38), rabbit (Broad/oryCun2), opossum (Broad/monDom5), chicken (ICGSC Gallus_gallus-4.0/galGal4), zebrafish (Zv9/danRer7), and elephant shark (Callorinchus_milii-6.1.3/calMil1). *Gcna* orthologs are indicated in green.

The following figure supplements are available for figure 3:

**Figure supplement 1.** Loss of structured domains in the murine lineage.

**Figure supplement 2.** Alignment of vertebrate GCNA proteins.

GCNA orthologs, allowing us to unambiguously identify such orthologs in a diverse array of eukaryotic species, more than two hundred in all (*Figure 4—source data 1*).

Interestingly, even though we discovered GCNA orthologs using alignable structured domains, our amino acid charge/hydropathy analysis reveals that, almost invariably, the proteins have a large non-alignable IDR at their N-terminus (*Figure 2A,B*). Despite a lack of primary sequence conservation, the overall amino acid composition of IDRs is often conserved in orthologous disordered protein families, as it is not the exact sequence of amino acids, but the overall character of the IDR – such as an abundance of residues that may be post-translationally modified, net charge, and amino acid enrichments and depletions – that confers function (*van der Lee et al., 2014*). Indeed, a comparison of mouse and human GCNA proteins showed that, while they were not readily aligned using conventional approaches, they do share highly repetitive and acidic IDRs (*Figure 1D*) (*Nolte et al., 2001*). A survey of the IDR composition of over 100 GCNA proteins across eukarya revealed that GCNA IDRs have characteristic amino acid enrichments and depletions – some shared with proteins in a disordered dataset (DisProt) and others specific to the GCNA family (*Figure 2C*). In keeping with another common IDR paradigm, GCNA proteins across species, including mouse and human, share short, linear protein-protein interaction motifs embedded within a larger disordered context (*Figure 1D*, *Figure 3—figure supplement 2*) (*van der Lee et al., 2014*; *Davey et al., 2015*).

GCNA orthologs are present in every eukaryotic superkingdom (*Figure 4A*) (*Simpson and Roger, 2004*). Although our gene discovery is enriched for animal and fungal orthologs due to the availability of sequenced genomes and transcriptomes, GCNA orthologs are present in the deepest known branches of eukaryotes. With certain exceptions including plants and protostomes such as drosophilids, each species contains one *Gcna* gene. Our analyses of these orthologs allowed us to deduce that a typical GCNA protein has four components: a large IDR, a zinc metalloprotease domain, a C2C2 zinc finger, and a non-canonical two-helix HMG box (*Figure 4B*, *Figure 4—figure supplement 1*). This four-domain architecture is generally conserved in GCNA proteins, although in some

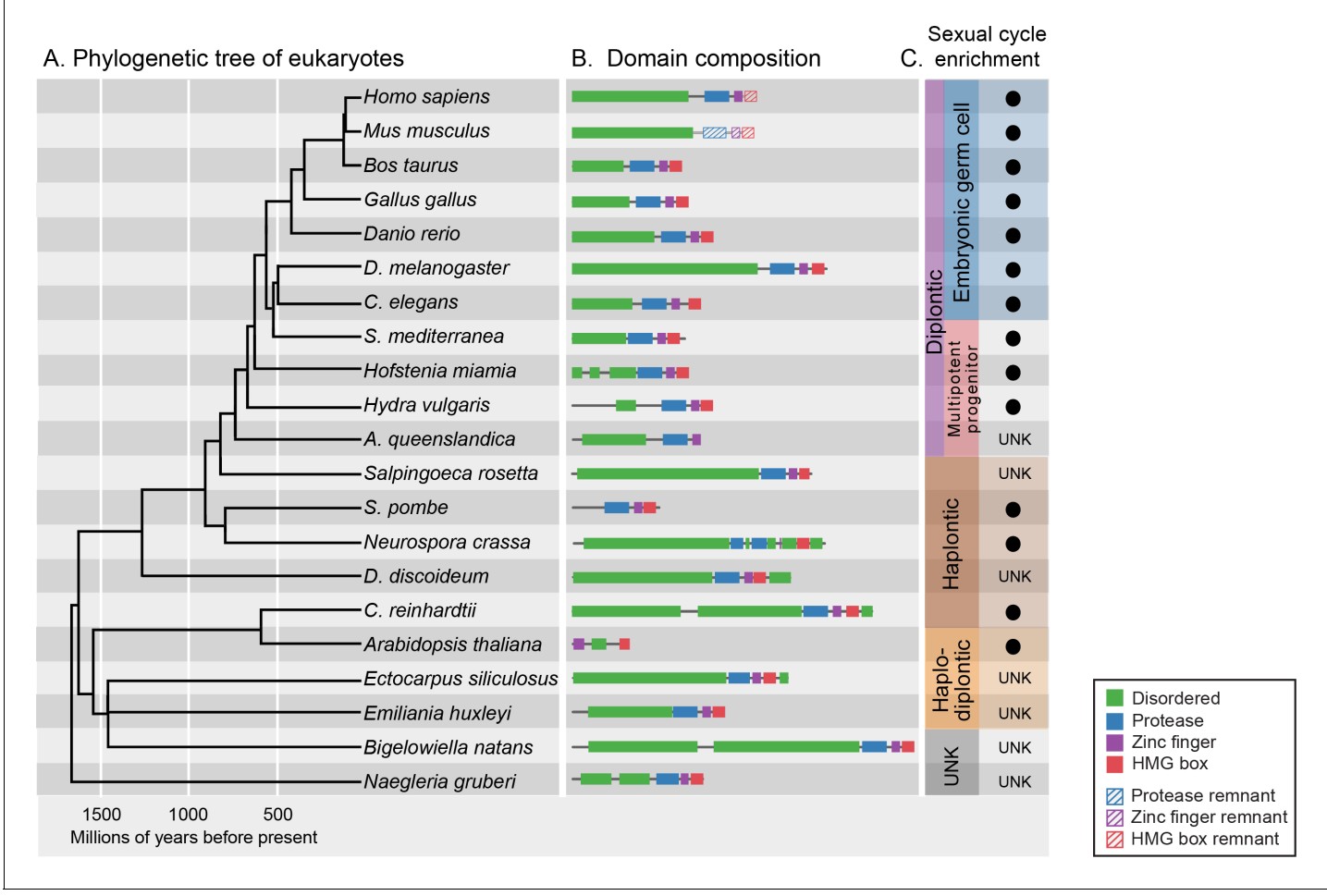

**Figure 4.** GCNA proteins have origins in the earliest eukaryotes and are associated with reproduction. (**A**) Phylogenetic tree of eukaryotes (topology of the tree is drawn as in [*He et al., 2014*], with estimated divergence times obtained from the TimeTree database [*Hedges et al., 2006*]). GCNA proteins are present in all major groups of eukaryotes regardless of the nature of their sexual life cycle – whether haplontic (haploid with a diploid phase that undergoes meiosis), diplontic (a diploid that undergoes meiosis to produce gametes), or haplodiplontic (alternates between haploid and diploid phases of the life cycle). (**B**) Domain composition of GCNA orthologs. Of note, *S. pombe* does not have an IDR, but a related fission yeast, *S. japonicus*, does (not shown). (**C**) Black circles indicate a given GCNA ortholog has enriched expression in reproductive cells or tissues or is upregulated during the sexual cycle. Organisms for which expression data do not exist are indicated by UNK (unknown).

The following source data and figure supplements are available for figure 4:

**Source data 1.** List of GCNA family members.

**Source data 2.** Reproductive expression of GCNA family members.

**Figure supplement 1.** Alignments and probabilistic hidden Markov models (HMMs) of structured domains found in GCNA family members.

**Figure supplement 1—source data 1.** Hidden Markov models used to discover GCNA orthologs.

**Figure supplement 2.** GCNA proteins in higher plants comprise the YABBY family.

**Figure supplement 3.** GCNA and YABBY HMG boxes are more closely related to each other than they are to any other type of HMG box, either within plants or across eukarya.

species, including mouse, one or more domains have been lost through modular protein evolution, bringing remarkable variation to the GCNA family (*Figure 4B*, *Figure 4—figure supplements 2* and *3*) (*Moore et al., 2013*). Taken together, our analyses lead us to infer that GCNA proteins were present in the common ancestor of eukaryotes, more than 2 billion years ago.

Of particular note are the GCNA proteins found in higher plants, where the protease domain has been lost. The YABBY proteins, a higher-plant-specific family whose evolutionary origins have long been sought (*Bartholmes et al., 2012*), share with other GCNA proteins the combination of a C2C2 zinc finger, an IDR, and a characteristic two-helix HMG box (*Figure 4—figure supplement 2*) (*Bowman and Smyth, 1999*). Based on our phylogenetic analyses of HMG boxes (*Figure 4—figure supplement 3*), we propose that YABBY proteins are GCNA family members that have diverged in the 600 million years since the last common ancestor of algae (where canonical four-domain GCNA proteins are still found) and higher plants.

## Across eukarya, GCNA is enriched in cells carrying the heritable genome

Since GCNA is a highly specific marker of both premeiotic and meiotic germ cells in the mouse, we sought to determine whether GCNA is associated with cells carrying the heritable genome either before or during the sexual cycle in other species by examining expression of GCNA homologs across eukarya, including fungi, plants, and the most basal single-celled eukaryotes (*Figure 4—source data 2*).

*Gcna* is expressed during the sexual cycle in single-celled haploid fungi and green algae, organisms that are more than a billion years removed from each other but share a similar life cycle. In *S. pombe*, transcription of *Gcna* is 17-fold upregulated during meiosis relative to vegetative cells (*Mata et al., 2002*). Likewise, in *Chlamydomonas,* a single-celled green alga, *Gcna* is upregulated 17-fold in activated gametes relative to vegetative cells (*Ning et al., 2013*).

In animals that generate germ cells from adult multipotent cells, *Gcna* is expressed more broadly in two cell types that have gametogenic potential and thus carry the heritable genome – the multipotent stem cells and the germ cells derived from them. In the cnidarian *Hydra*, germ cells are derived from interstitial stem cells, where *Gcna* expression is upregulated compared with two exclusively somatic stem cell lineages (*Littlefield, 1985*; *Bosch et al., 2010*; *Hemmrich et al., 2012*). In the planarian flatworm *Schmidtea mediterranea*, and the acoel *Hofstenia miamia*, *Gcna* is upregulated in neoblasts, the totipotent adult stem cell populations from which germ cells arise (*van Wolfswinkel et al., 2014*; *Srivastava et al., 2014*). Transcripts of this gene are also enriched in the testes in a sexual strain of *Schmidtea* (*Figure 5A*).

We also found enrichment of *Gcna* in germ cells of animals that specify their germline during embryonic development (*Extavour and Akam, 2003*). *Gcna* is enriched in the germlines of animals such as *C. elegans, Drosophila,* zebrafish, and frog, which segregate their germlines in the first cell division of the embryo due to maternally inherited factors (*Figure 5B,C*, *Figure 5—figure supplement 1*, *Figure 4—source data 2*) (*Tomancak et al., 2002*; *Graveley et al., 2011*; *Reinke et al., 2004*; *Robinson et al., 2013*; *Kohara and Shin-i, 1998*). Likewise, mouse, which specifies germ cells later in the embryo through inductive signals, shows a 13-fold upregulation of *Gcna* transcript in the testis compared to somatic tissues (*Figure 4—source data 2*). Transcripts in somatic tissues are apparently not translated, as mouse GCNA protein is germ-cell specific (*Maatouk et al., 2006*). An array of other mammals shows similar upregulation in the testis (*Figure 5—figure supplement 1*). Like mouse GCNA protein, the human ortholog is localized to the nucleus of germ cells (*Figure 5D, E*).

In summary, we found that expression of *Gcna* genes is enriched in cells carrying a heritable genome – either just prior to or during meiosis in single-celled eukaryotes, in stem cells and germ cells of animals without a dedicated germline, and in germ cells of organisms with a dedicated germline (*Figure 4C*, *Figure 4—source data 2*). *Gcna* expression across all of these modalities indicates that *Gcna* was present, and was likely enriched in cells carrying a heritable genome, in the last common ancestor of all extant eukaryotes.

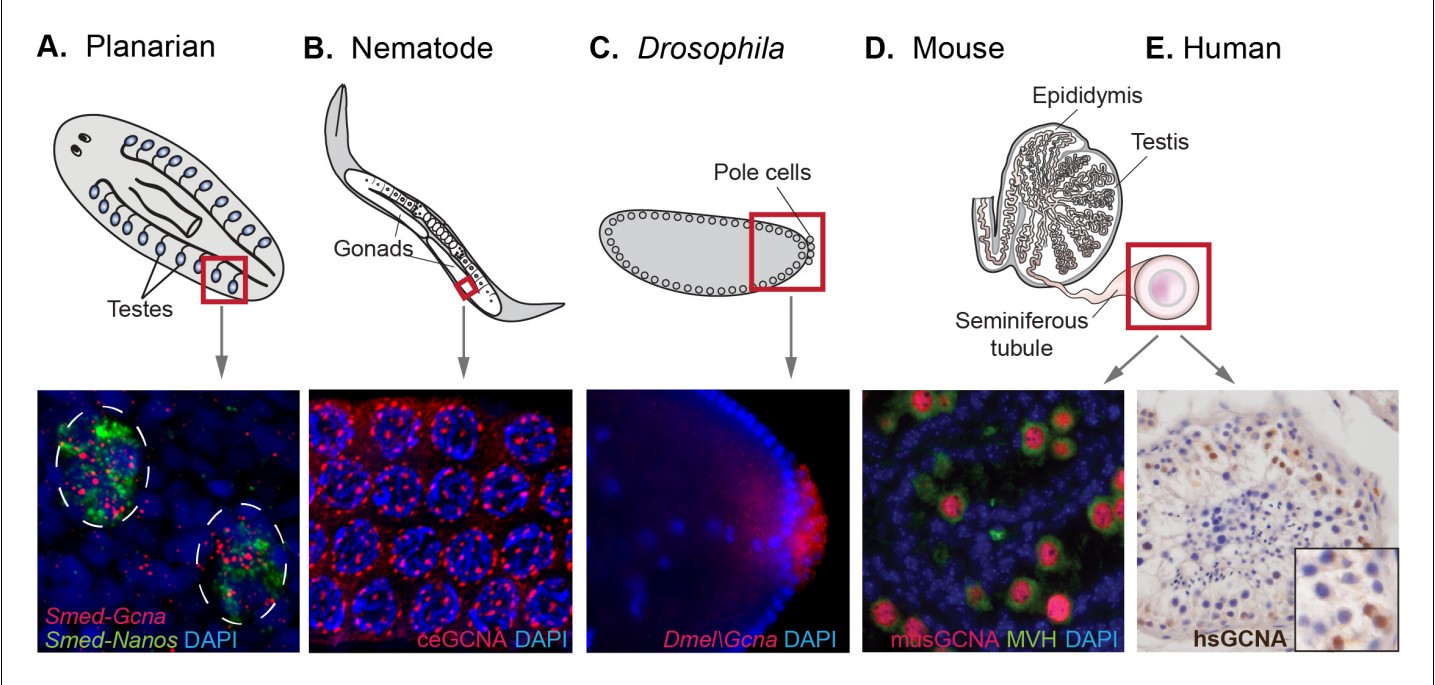

**Figure 5.** GCNA RNA and protein are enriched in germ cells of a variety of model organisms. (**A**) Germ cells residing in testes (circled) of a sexual strain of the planarian flatworm *Schmidtea mediterranea* are enriched for *Gcna* transcript (red), as well as that of germ cell marker *Nanos* (green). (**B**) Germ cells in pachytene stage of meiosis in nematode *C. elegans* express GCNA protein (red). Nuclei stained with DAPI (blue). (**C**) Pole cells, the earliest germ cells in a *Drosophila melanogaster* embryo, are enriched for transcript of *Gcna* ortholog CG14814. Image reproduced with permission (*Lécuyer et al., 2007*). GCNA protein is expressed in germ cells in cross sections of mouse (**D**) and human (**E**) seminiferous tubules, the site of meiosis and sperm production. (**D**) Pre-meiotic germ cells in testis of 1-day old mouse are labeled using antibodies recognizing GCNA (red) and DDX4/MVH (green). (**E**) Antibody recognizing human GCNA labels nuclei of germ cells (brown). Red squares indicate origin of structures and/or cells depicted below each diagram.

The following figure supplement is available for figure 5:

**Figure supplement 1.** Enrichment of *Gcna* expression in reproductive tissues of vertebrates.

## *C. elegans gcna-1* mutants present reproductive phenotypes

Given the extraordinary association of *Gcna* expression with reproductive cells and tissues across eukarya, we set out to test whether this expression translates to conservation of reproductive function. The only GCNA family member whose function had been probed previously is the *C. elegans* gene ZK328.4, which we have named *gcna-1*. *C. elegans gcna-1* had been depleted as part of an RNAi screen of germline-enriched genes (*Colaiácovo et al., 2002*). *gcna-1* was reported to have a low penetrance HIM (<u>H</u>igh <u>I</u>ncidence of <u>M</u>ales) phenotype, suggestive of a defect in meiotic chromosome segregation, as XO male progeny result from nondisjunction in XX hermaphrodites. We decided to further investigate *gcna-1* function in *C. elegans* by creating two independent mutant alleles (*Figure 6—figure supplement 1*). Worms carrying either of the *gcna-1* mutant alleles have small but significant reductions in brood size at 25 degrees Celsius (*Figure 6A*). Additionally, both alleles exhibit a significant HIM phenotype, substantiating a reproductive function for GCNA in *C. elegans* (*Figure 6B*).

C. elegans *gcna-1* encodes a canonical GCNA protein with an IDR, protease domain, zinc finger, and HMG box. Even in the face of other domains being lost over evolutionary time, the large IDR is retained in GCNA proteins almost without exception. To understand the persistence of this rapidly evolving domain across billions of years, we sought to probe the functional role of the IDR in GCNA proteins. Mouse GCNA presented us with the unique opportunity to probe the function of the IDR in isolation, as it has an IDR that has existed for 25 million years in the absence of structured domains.

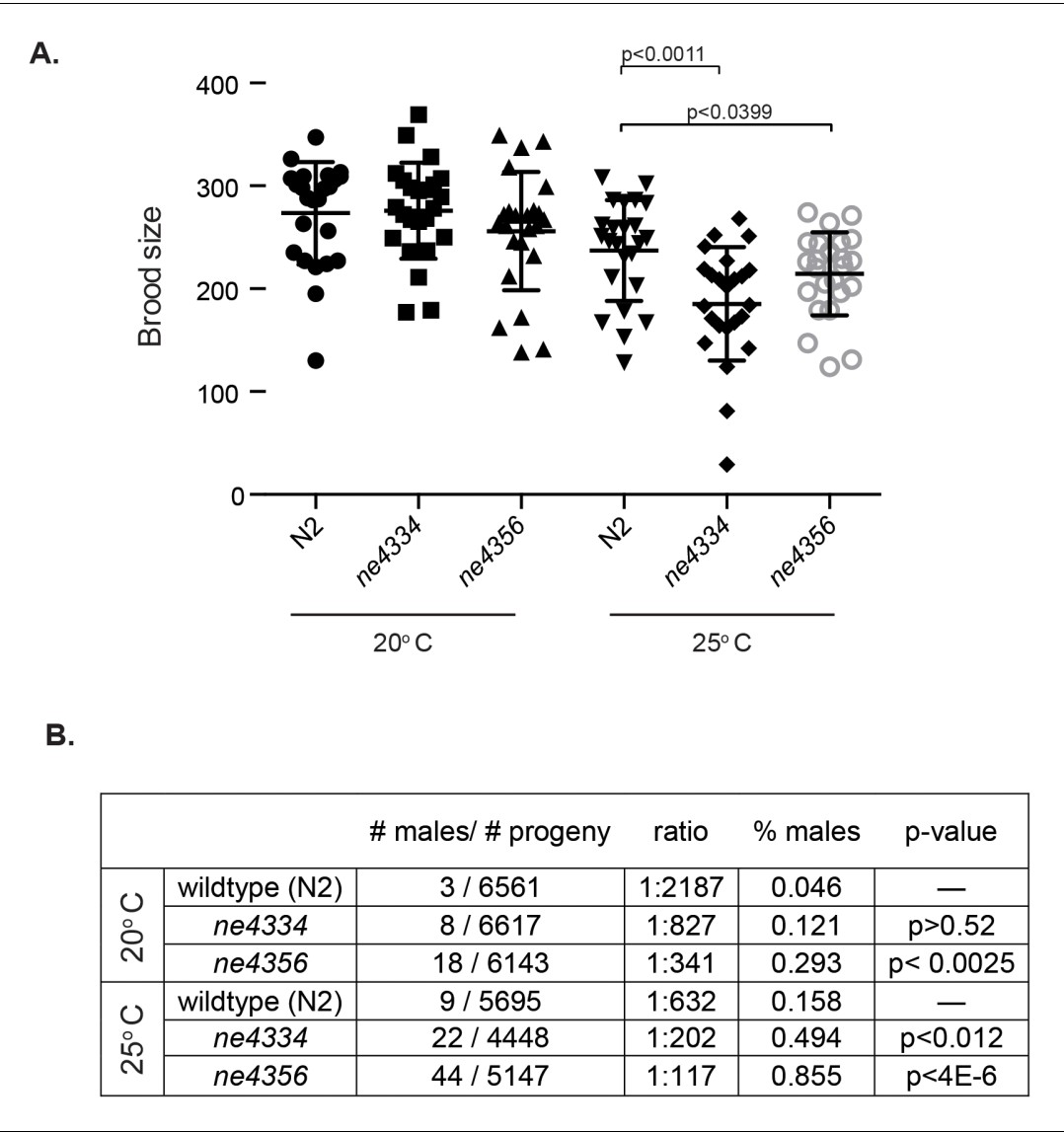

**Figure 6.** *C. elegans gcna-1* mutants present reproductive phenotypes. (**A**) Two alleles of *gcna-1* in *C. elegans* exhibit significant reduction in brood size at 25 degrees Celsius. Twenty-four broods were counted for each allele at each temperature. Each symbol represents the number of progeny derived from a single hermaphrodite. Outliers are shown and no data was excluded. Bars indicate mean +/- standard deviation. P-values were calculated with two-tailed non-parametric Mann-Whitney test. (**B**) *gcna-1* mutants also exhibit a high incidence of male progeny at both 20 and 25 degrees Celsius. Twenty-four broods were counted and scored for each allele at each temperature. P-values were calculated using a chi-squared test with correction for multiple testing.

The following figure supplement is available for figure 6:

**Figure supplement 1.** *C. elegans gcna-1* alleles.

## The entirely disordered mouse GCNA is required for male fertility

To this end, we created a targeted conditional allele of *Gcna* in the mouse (***Figure 1—figure supplement 2***). We found that mutating *Gcna* dramatically impairs male fertility, a testament to the importance of the IDR; 10 of 11 *Gcna*-mutant males tested sired no offspring, while wild-type male controls sired 5 to 24 offspring (***Figure 7A***). The epididymal duct, which contains copious amounts of maturing sperm in wildtype mice, was nearly devoid of sperm in the mutant (***Figure 7B***). *Gcna-*

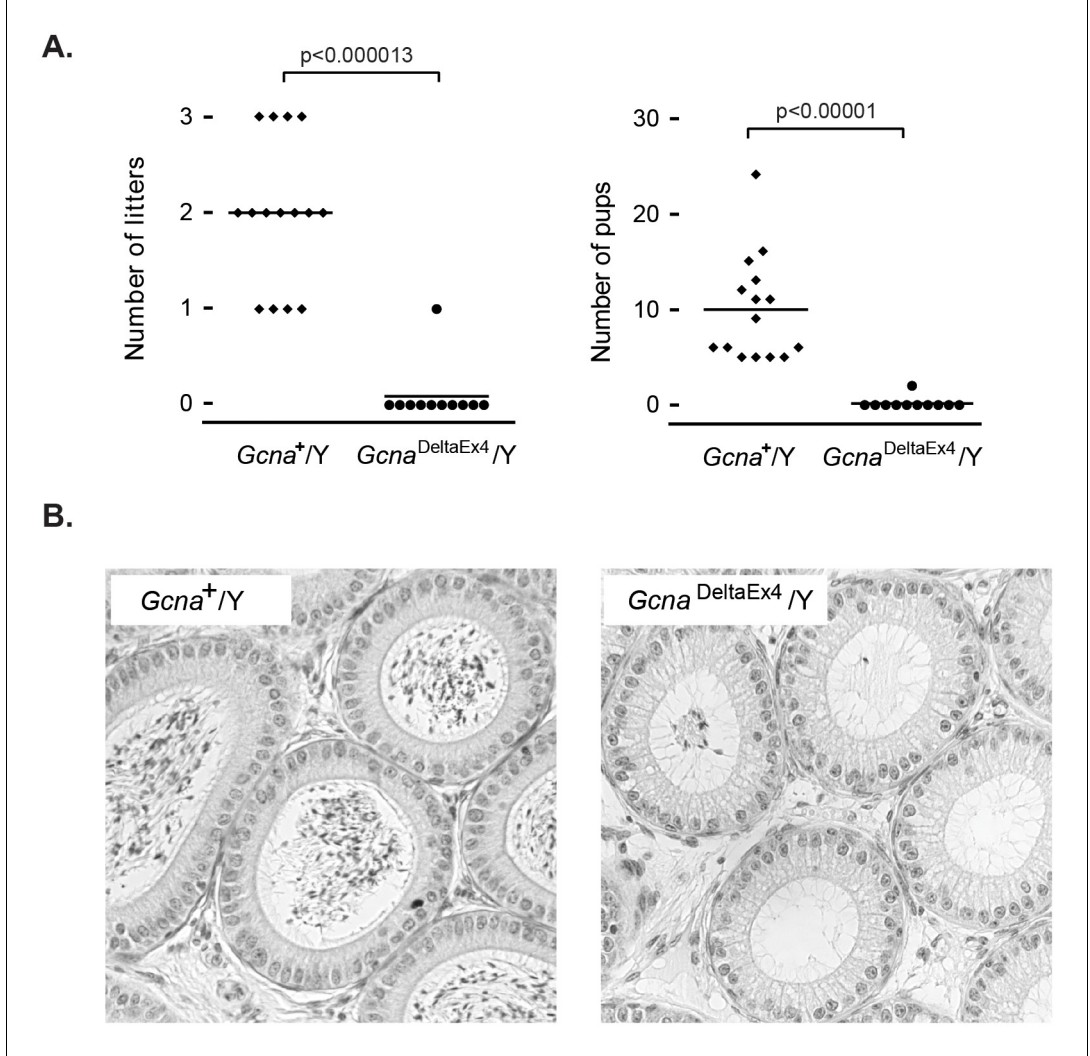

**Figure 7.** *Gcna* is required for male fertility in mice. (**A**) *Gcna*-mutant male mice exhibit marked subfertility, and most are sterile. Each datapoint represents the number of litters (**A**) and pups (**B**) sired by a single male (n=15 for wildtype, n=11 for *Gcna*DeltaEx4/Y). P-values were calculated using a two-sided Wilcoxon rank sum test. (**B**) Epididymal ducts in *Gcna*-mutant males are largely devoid of sperm in the lumen when compared to those of wildtype mice.

mutant female mice were fertile (data not shown). Sex-specific infertility is a common phenomenon in mice, and is thought to be due to different levels of checkpoint control in males and females (*Matzuk and Lamb, 2002*; *Morelli and Cohen, 2005*; *Hunt and Hassold, 2002*).

## GCNA is homologous to two protein families involved in replication-associated DNA repair

To gain insight into potential molecular functions of GCNA that might underlie these phenotypes, we extended our molecular evolution analyses beyond the GCNA family to other proteins with similar domain composition. Through phylogenetic analysis as well as structural modeling, we determined that GCNA proteins are members of a small family of IDR-containing metalloproteases that includes only two other members, Wss1 and Spartan, both of which have essential functions in DNA repair coupled to DNA replication (*Mosbech et al., 2012*; *Centore et al., 2012*; *Davis et al., 2012*; *Juhasz et al., 2012*; *Machida et al., 2012*; *Kim et al., 2013*; *Stingele et al., 2014*; *Balakirev et al., 2015*). GCNA, Wss1, and Spartan proteins share three common elements: minigluzincin protease

domains, large intrinsically disordered regions (IDRs), and motifs for binding ubiquitin family proteins (particularly SUMO interacting motifs, or SIMs).

Wss1 and Spartan proteins have recently been recognized as members of the same protease family (*Stingele et al., 2015*). Our phylogenetic analyses based on amino acid sequence alignments of Wss1, Spartan, and GCNA protease domains show that GCNA proteases are also part of this small and distinctive group (*Figure 8A*). Secondary structure prediction and 3D modeling place all three proteins within the minigluzincin protease family (*Figure 8B*) (*Balakirev et al., 2015*; *López-Pelegrín et al., 2013*). We conducted HMM to HMM comparison across more than 14,800 PfamA curated protein families, including over 200 protease families, using alignment and secondary structure scoring. We found that GCNA, Spartan, and Wss1 proteases are more closely related to each other than they are to any other protein family (*Figure 8—figure supplement 1*).

Cementing the relationship between these three proteins is a second common element shared by GCNA, Wss1, and Spartan – a large disordered domain. Like those in GCNA, the Wss1 and Spartan IDRs are present across a large swath of eukarya (*Figure 9A*). IDRs are well known to function in the flexible display of short linear motifs that are required for protein-protein interactions (*van der Lee et al., 2014*). These short motifs represent small islands of conservation within IDRs. Among these short motifs, SUMO interacting motifs (SIMs) constitute the third common element shared by GCNA, Wss1, and Spartan (*Figure 9B*, *Figure 3—figure supplement 2*).

## Discussion

In summary, we find that an antigen widely employed to identify germ cells in mice corresponds to a protein that has been expressed for nearly 2 billion years in cells carrying a heritable genome – whether single cells during meiosis, stem cells that give rise to germ cells, or germ cells themselves. We provide evidence that GCNA functions in reproduction in two organisms – *C. elegans* and mouse – whose last common ancestor lived about 600 million years ago (*Cartwright and Collins, 2007*; *Parfrey et al., 2011*; *Peterson et al., 2008*), and suggest that GCNA may have a function in cells carrying a heritable genome across eukarya.

The GCNA protein features, and in mice consists entirely of, an intrinsicially disordered region with an unusual and rapidly evolving amino acid sequence that has kept this protein hidden from researchers until now. Protein disorder is ubiquitously present in eukaryotic proteomes; greater than 30% of proteins have disordered segments of 30 or more consecutive residues (*Dunker et al., 2000*). IDRs, rather than having a single, well-defined fold, dynamically interconvert between an ensemble of conformations separated by low energy barriers (*Chebaro et al., 2015*). IDRs therefore challenge the expectations set forth in the structure-function paradigm, which posits that the function of a protein is dependent upon its three-dimensional structure (*Wright and Dyson, 1999*). IDRs have critical biological functions across a wide variety of cellular processes including transcription, regulatory, and signaling pathways, and it has been proposed that IDRs support the processes that underlie the development of multicellular organisms (*Xie et al., 2007*; *Tantos et al., 2012*; *Dunker et al., 2015*).

Male sterility caused by mutation of the entirely disordered mouse GCNA indicates that the function of GCNA proteins lies at least in part in the disordered region. While the molecular functions of GCNA IDRs remain to be elucidated, prior studies of other IDRs (*van der Lee et al., 2014*) and specifically the known roles of the IDRs in the homologous proteins Wss1 and Spartan inform our thinking about GCNA IDR functions. Homology with Wss1 and Spartan proteins suggests that GCNA IDRs function in molecular recognition mediated by SUMO interacting motifs, thereby recruiting or scaffolding larger protein complexes. GCNA IDRs may also contribute to nucleic acid binding, either alone or in combination with the zinc finger and HMG box.

GCNA may be among the first of many disordered protein families to be implicated in reproduction, as sex chromosomes, which have been shown to be enriched for genes encoding germ cell genes, are also enriched for disordered proteins (*Mueller et al., 2013*; *Hegyi and Tompa, 2012*). Spermatogenesis and meiosis are in fact among the biological processes most strongly correlated with predicted protein disorder (*Dunker et al., 2015*).

Conservation of structured domains in the GCNA family over a vast evolutionary timescale suggests that significant function likely also derives from these domains. We have identified both sequence and structural homology to Wss1 and Spartan, encompassing the protease domain, IDR,

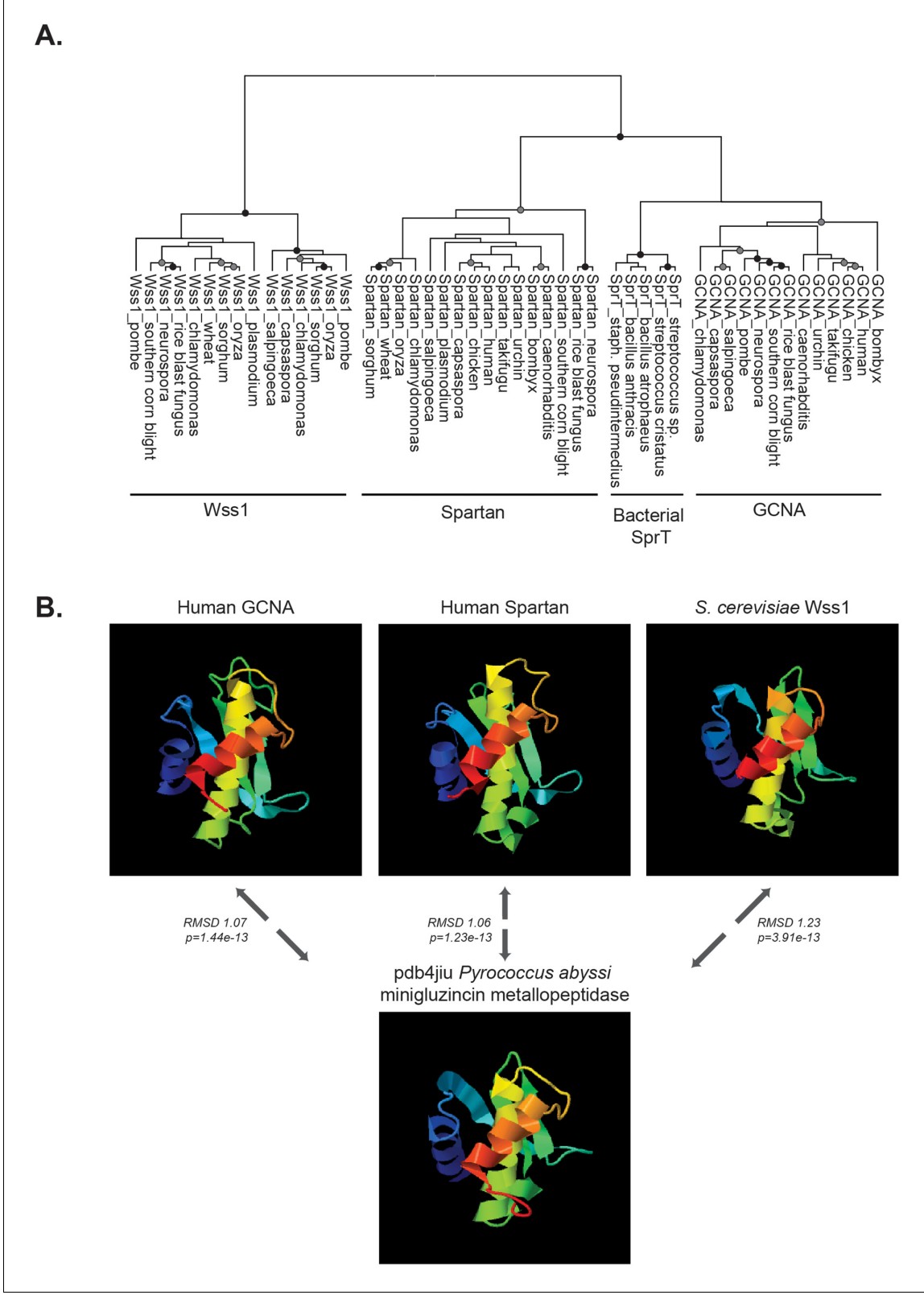

**Figure 8.** GCNA, Wss1, and Spartan proteins comprise a family of proteases. (**A**) Maximum likelihood phylogenetic tree showing relationships among eukaryotic GCNA, Wss1, Spartan, and bacterial SprT protease domains. GCNA, Wss1, and Spartan are present across eukarya, including the most primitive eukaryotes, except that Wss1 proteins have been lost in animals. Gray and black circles indicate nodes with bootstrap values greater than 600 and 900 (out of 1000), respectively. Tree is based on alignment of the protease domains and was created with PhyML. (**B**) Structural modeling of GCNA,
*Figure 8 continued on next page*

*Figure 8 continued*

Wss1, and Spartan protease domains places GCNA in the minigluzincin family of proteases along with Wss1 and Spartan. Pairwise protein structure comparison using FATCAT (*Ye and Godzik, 2004*) detected strong structural similarity, as evidenced by small root-mean-square deviations (RMSDs), which are measures of the average distance (in Angstroms) between the carbon atoms of the superimposed proteins. All structures were found to be significantly similar.

The following figure supplement is available for figure 8:

**Figure supplement 1.** GCNA, Spartan, and Wss1 form a subgroup within the larger Peptidase Clan MA.

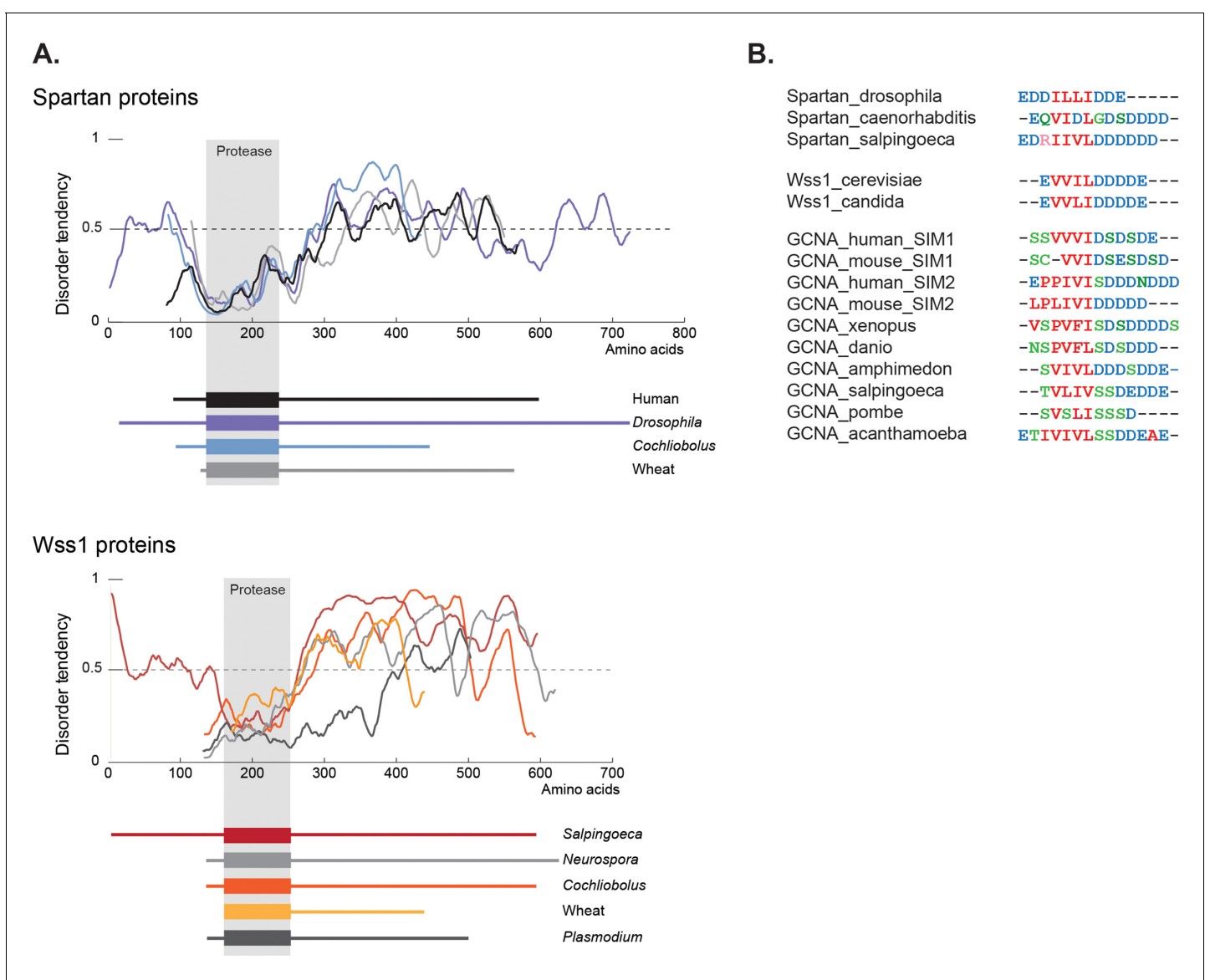

**Figure 9.** GCNA, Wss1, and Spartan proteins have SUMO interacting motifs within large disordered regions. (**A**) Spartan and Wss1 have significant disordered regions outside of their protease domains. Residues above the dotted line are predicted to be disordered. (**B**) GCNA proteins contain conserved SUMO-interacting motifs (SIMs) that are homologous to those found in Spartan and Wss1 proteins. Residues are colored as follows: blue (acidic), green (hydroxyl, sulfhydryl, amine), red (small and hydrophobic), pink (basic).

and SUMO interacting motifs. Wss1 and Spartan protease domains and SUMO interacting motifs are essential for response to DNA damage caused by ionizing radiation, UV light, and chemical cross-linkers (*Centore et al., 2012*; *Stingele et al., 2014*). GCNA shares these common elements. By extrapolation, we propose that the GCNA protein family identified here may also function in maintenance of genome integrity, and has become specialized for cells whose genomes will be passed to the next generation. As such, GCNA may help cells cope with the additional DNA damage burden that accompanies meiosis, a process that involves extensive, programmed DNA damage (*Keeney, 2008*).

While at first glance the phenotypes of mouse and *C. elegans Gcna* mutants seem disparate, they are consistent with defects in DNA damage repair in both organisms. The phenotype of *Gcna* mutant mice is similar to those of other genes involved in DNA damage repair, including *Rad18* and *Rad6*, which function in the same DNA damage tolerance pathway as Spartan (*Inagaki, 2011*; *Roest et al., 1996*; *Hedglin and Benkovic, 2015*). Likewise, the HIM phenotype in *Gcna*-deficient *C. elegans* is consistent with a defect in genome maintenance (*Billi et al., 2014*); in *C. elegans* as well as mammals, efficient and proper repair of meiotic double strand breaks is crucial for accurate chromosome segregation (*Baudat et al., 2013*; *Youds and Boulton, 2011*).

GCNA joins a well studied cohort of genes, including *Dazl/Boule, Vasa, Nanos*, and *Piwi*, known for their function in metazoan germ cells. Many of these genes are proposed to be part of a conserved germline multipotency program (GMP) that functions in multipotent cells as well as in the germline, where they have a broad role in establishing and maintaining pluripotency (*Ewen-Campen et al., 2010*; *Juliano et al., 2010*). Unlike most of these genes (*Eirín-López and Ausió, 2011*; *Kerner et al., 2011*), GCNA arose prior to the divergence of the major eukaryotic lineages. GCNA is more ancient than all but *Piwi* and associated proteins that function in small-RNA-mediated genome defense (*Kerner et al., 2011*; *Swarts et al., 2014*), and predates the origin of a dedicated metazoan germline by a billion years. We believe that GCNA's presence in the last common ancestor of extant eukaryotes indicates that GCNA's role is more fundamental than even the ancestral pluripotency module and the GMP that arose from it. Like *Piwi*, GCNA proteins may perform a role that is critical for reproductive cells by helping to maintain the integrity and stability of heritable genomes across the generations.

## Materials and methods

### Whole-mount immunofluorescence

E11.5 mouse embryos that carried an Oct4-EGFP transgene (C57BL/6N;CBA-Tg(Pou5f1-EGFP) 2Mnn/J (RRID:MGI:3581420), backcrossed at least 7 generations onto C57BL/6N) were dissected to remove heads, limbs, body walls, and internal organs. Whole-mount staining was carried out as described (*Hu et al., 2013*). Briefly, embryos were fixed overnight and then blocked for another night. After washes, embryos were incubated at 4°C overnight with antibodies against GATA4 (sc-25310, Santa Cruz Biotechnology, RRID:AB_627667), and GCNA1 (RRID:AB_2629436). After washes, embryos were then incubated at 4°C overnight with donkey secondary antibodies conjugated with Rhodamine Red X, or DyLight 649 (Jackson ImmunoResearch). Oct4-EGFP expression marks germ cells. All antibodies were diluted 1:100. After extensive washes, embryos were preserved in Slow-Fade Gold Antifade reagent (Life Technologies). Stained embryos were imaged sagittally using an LSM510 confocal microscope (Zeiss).

### Immunoprecipitation and mass spectrometry

One wildtype adult testis per IP was dounced in 1 ml IP buffer. IP buffer: 50 mM HEPES pH 7.4, 140 mM NaCl, 10% glycerol, 0.5% NP-40, 0.25% Triton, 100 uM ZnCl2 plus EDTA-free Protease Inhibitor Tabs (Roche). Samples were sonicated on ice 1 min at 30% amplitude using a Branson Sonifier and treated with 100 U/ml Benzonase (Millipore) for 20 min at room temperature on rocker. Samples were spun down for 10 min at 16,000 G at 4°C, and supernatant was used for immunoprecipitations. Ten milliliters of GCNA1 hybridoma (10D9G11) supernatant (RRID:AB_2629436) and 50 ug of Rat IgM isotype control (Southern Biotech 0120-01, RRID: AB_2629437) were coupled to 50 ul Goat-Anti-Rat IgM agarose (Open Biosystems, SAB1120) using dimethylpimelimidate (Sigma D8388). 25 ul of beads were used per IP. TRA98 IPs were carried out using 5 ug/ml TRA98 antibody

(Abcam, ab82527, RRID:AB_1659152) and 5 ug/ml Rat IgG isotype control (ChromPure Jackson Immunoresearch, RRID:AB_2337136) using Protein G Agarose (Upstate). After extensive washing in IP buffer, precipitated proteins were subjected to SDS–PAGE and silver staining. Samples were processed at the Whitehead Institute Proteomics Core Facility. For mass spectrometry analysis, bands were excised from each lane of a gel encompassing the entire molecular weight range. Trypsin digested samples were analyzed by reversed phase HPLC and a ThermoFisher LTQ linear ion trap mass spectrometer. Peptides were identified from the MS data using SEQUEST (RRID:SCR_014594).

## Isoelectric point calculations

Isoelectric points of all proteins in the mouse proteome (UniProtKB, version 2014-06-17, RRID:SCR_004426) were calculated using the Henderson–Hasselbalch equation. EMBOSS (RRID:SCR_008493) pK values were used to calculate isolectric points. Amino acid (pK): Amino (8.6), Carboxyl (3.6), C (8.5), D (3.9), E (4.1), H (6.5), K (10.8), R (12.5), Y (10.1) (*Rice et al., 2000*). Perl code is provided in *Source code 1*.

## ES cell targeting

Targeting vector was designed and built as in the KOMP (RRID:SCR_007318) pipeline (*Skarnes et al., 2011*) to ultimately generate an allele with Exon 4 of mouse *Gcna* flanked by LoxP sites. Mouse embryonic stem cells (E14Tg2a.4 derived from 129P2/OlaHsd, UC Davis, RRID: MMRRC_015890-UCD) were electroporated with the targeting vector and clones were selected using Neomycin. Homologous recombinants were identified by Southern blotting to confirm correct integration. Integration was evaluated after digestion with NheI and a probe generated with the primers 5'-TCAGTGCGGTGCAGTGATA-3' and 5'-CCAAGTTCCCACACAGAACC-3'. The 'knockout-first' allele contains an IRES:*lacZ* trapping cassette and a floxed promoter-driven *neo* cassette inserted into the intron between Exons 4 and 5. ES cells were electroporated with FLPe plasmid pCAGGS-FlpE-PURO (*Akbudak and Srivastava, 2011*) to convert the 'knockout-first' allele to a conditional allele. Clones were selected after transient puromycin selection and screened by PCR for the Flp event using the primers 5'-ATAAGAGAGGGAAAGGTTGTGGCTT-3' and 5'-TGATATCTCTATAG TCGCAGTAGGC-3'. The same targeting vector was electroporated into JM8 ES cells (RRID:CVCL_J957) to generate a *Gcna* targeted allele on a C57BL/6N background. Mice were bred to *Ddx4/ Mouse vasa homolog cre mice* (Mvh^Cre-mOrange) (*Hu et al., 2013*) to delete exon 4 *in vivo*.

## Evaluation of GCNA protein in mouse embryonic stem (ES) cells

GCNA conditional (Exon 4 floxed) ES cells were electroporated with a pCAGGS-CRE plasmid, plated at clonal density, and clones screened by PCR for the Exon 4 deletion event. ES cells were harvested using a cell scraper and RIPA lysis buffer. After lysing, debris was spun down and supernatant was run on an SDS/PAGE gel. Western blots on Exon 4 floxed and deleted ES cells were carried out with TRA98 (Abcam ab82527, 1:500, RRID:AB_1659152) and GCNA1 (Enders, 1:20, RRID:AB_2629436) antibodies.

## Bacterial expression of GCNA

GCNA cDNA was amplified from adult mouse testis RNA and cloned into pENTR-D/TOPO (Life Technologies). Primer sequences for N-terminal portion of protein were: N-term-for (5'-CACCC TGAACATGGATTCAGGC-3') and N-term-rev (5'-GCTCAAGCTCACTTGATGT-3'). Primers for repetitive portion of the protein were: Repeats-for (5'-CACCACATCAAGTGAGCTTGAGC-3') and Repeats-rev (5'-TTTTTGGCTTCCTCAGCTGT-3'). N-terminal and GE(P/M/S)E(S/T)EAK repeat encoding portions of the GCNA cDNA were transferred into pDEST17 Gateway vector containing an N-terminal 6X HIS tag using LR Clonase enzyme (Thermo Fisher). Fusion proteins were expressed in ROSETTA 2(DE3) pLysS bacteria (MerckMillipore). Bacteria were harvested 1 hr after induction with 1 mM IPTG, centrifuged, lysed in SDS-PAGE loading buffer, and sonicated before loading onto an SDS/PAGE gel. Western blots were carried out with TRA98 (Abcam ab82527, 1:500, RRID:AB_ 1659152), GCNA1 (Enders, 1:20, RRID:AB_2629436), and anti-penta-HIS (Qiagen, 1:5000, RRID:AB_ 2619735) antibodies.

## Disorder plot

Disorder was predicted using the IUPRED web server (RRID:SCR_014632) with default parameters (*Dosztányi et al., 2005*). Data was smoothed using exponential smoothing with a damping factor of 0.9 before plotting in Microsoft Excel.

## Alignments

Alignments were carried out using MUSCLE (RRID:SCR_011812) (*Edgar, 2004*), manually adjusted in Jalview (RRID:SCR_006459) (*Waterhouse et al., 2009*), and shaded using Boxshade (RRID:SCR_011812) (http://www.ch.embnet.org/software/BOX_form.html).

## GCNA protein identification

Profile HMMs for the protease domain, zinc finger, and HMG box were constructed using the HMMbuild function of the command line version of the HMMER3 software suite (RRID:SCR_005305) (*Eddy, 2009*) with default settings. GCNA proteins from the following species were used to construct the protease domain and zinc finger HMMs: *Homo sapiens* (ACRC_HUMAN), *Bos taurus* (XP_005228100.1), *Gallus gallus* (XP_420196.3), *Danio rerio* (AAI54363), *Drosophila melanogaster* (CG14814), *Caenorhabditis elegans* (ZK328.4), *Hydra vulgaris* (T2M7F9_HYDVU), *Monosiga brevicollis* (XP_001744295), *Schizosaccharomyces pombe* (YGM4_SCHPO), *Schizosaccharomyces japonicus* (B6K1W0_SCHJY), *Ostreococcus tauri* (XP_003074438.1), *Emiliania huxleyi* (R1E9K5_EMIHU), *Chlamydomonas reinhardtii* (Cre01.g003700), and *Dictyostelium discoideum* (Q54JE2_DICDI). As human GCNA has an in-frame stop codon before its HMG box, the HMG box of *Tupaia chinensis,* the mammal closest to primates, was used in the HMG box HMM construction. HMM logos were generated using the web server at http://skylign.org (RRID:SCR_001176) (*Wheeler et al., 2014*). HMMSearch (*Finn et al., 2015*) was used to query the UniProt database (UniProtKB, version 2015_03, RRID:SCR_004426) with the profile HMMs. Hits with an E-value smaller than <1E-05 were considered significant. Duplicate UniProt entries were filtered out using reciprocal BLAST (RRID:SCR_004870). For proteins with several predicted isoforms, the longest isoform was kept. For sequences from strains of the same species, only one strain was included in the master list. Using these criteria, we identified 160 UniProt entries containing significant matches to HMMs from all three domains. This number of orthologs is likely to be an underestimate of the number of GCNA proteins across eukarya. For example, we found an additional 136 UniProt entries that contained significant matches to only the protease domain and zinc finger. This number includes primate GCNAs which have an in-frame stop codon before the HMG box. It is unclear whether other species outside of primates are indeed missing the HMG box, as many of these proteins are predictions that may suffer from poor annotation. For those species not well-represented in UniProt, we searched the Joint Genome Institute's data by BLAST at http://genome.jgi.doe.gov/ (RRID:SCR_002383). A listing of GCNA proteins can be found in *Figure 4—source data 1*.

## Phylogenetic tree construction

Alignments were generated using MUSCLE (RRID:SCR_011812) (*Edgar, 2004*). ProtTest (RRID:SCR_014628) was used to select the best-fit model for protein evolution (*Abascal et al., 2005*). PhyML (RRID:SCR_014629) version 20111216 was used to create maximum likelihood trees (*Guindon et al., 2010*). Command line used was as follows: phyml -i phylipalignment.txt -d aa -b 1000 -m LG -f m -v e -a e -s BEST -o tlr.

## Human immunohistochemistry

Normal human testis paraffin sections were purchased from IHC World, LLC (TS-H5023). After microwave-assisted antigen retrieval in Antigen Retrieval Buffer 1 (Spring Bioscience PMB1-250), endogenous peroxidases were quenched in 3% $H_2O_2$ for 10 min. Slides were blocked in 3% BSA and incubated with anti-ACRC (human GCNA) antibody (HPA023476, RRID:AB_1844530) at 1:20 dilution in 1% BSA in PBS. After washing in PBS, slides were incubated with ImmPress HRP Anti-Rabbit Ig (RRID:AB_2336533) and developed using ImmPact DAB peroxidase substrate (Vector Laboratories). Slides were stained with hematoxylin and mounted in Permount.

## Schmidtea in situ hybridization

Planarian FISH was performed as previously described (*King and Newmark, 2013*). Briefly, hatchlings of the S2 sexual strain were fixed in 4% formaldehyde, bleached in formamide bleaching solution (5% formamide, 0.5xSSC, 1.2% $H_2O_2$), and permeabilized by Proteinase K treatment (2 µg/ml in PBS with 0.1% SDS). Riboprobes containing DIG-12-UTP (Roche) or DNP-11-UTP (PerkinElmer) were generated by *in vitro* transcription from custom PCR products of the entire open reading frame with an attached T7 promoter. Hybridizations were performed overnight at 56°C in hybridization buffer (50% formamide, 5xSSC, 5% dextran sulphate, 1% Tween-20, and 20 µg/ml yeast RNA). Probes were detected with peroxidase-conjugated anti-DIG antibody (RRID:AB_514496, Roche/Sigma-Aldrich) or anti-DNP (RRID:AB_2629439, PerkinElmer) and developed by Tyramide Signal Amplification (TSA) in TSA buffer (100 mM borate pH 8.5, 2 M NaCl, 0.003% $H_2O_2$, and 20 µg/ml 4-iodophenylboronic acid). Tyramide-conjugated fluorophores were generated from fluorescein and rhodamine N-hydroxysuccinimide (NHS) esters (Pierce) as previously described (*Hopman et al., 1998*). Between developments, residual peroxidase activity was quenched by incubation in 1% sodium azide. Samples were counterstained with DAPI (Sigma; 1 µg/ml). Images were acquired on a Zeiss LSM700 confocal microscope.

## *C. elegans* antibody generation and immunohistochemistry

A rabbit polyclonal antibody against a peptide in the disordered region of *C. elegans gcna-1* (ZK328.4) (YPEMFDSNQKPRQKPKE) was generated by Yenzym Antibodies, LLC. Antiserum was affinity-purified using SulfoLink (Pierce) following the manufacturer's instructions. Whole mount preparations of dissected gonads, fixation, and immunostaining procedures were carried out as described in (*Colaiácovo et al., 2003*). Primary antibody was used at 1:100 dilution. Images were collected using a DeltaVision system (Applied Precision) and subjected to deconvolution and projection using the SoftWoRx 3.3.6 software (Applied Precision).

## C. *elegans* strains and genetics

The N2 Bristol strain of *C. elegans* was cultured at 20°C under standard conditions as described in Brenner (*Brenner, 1974*). CRISPR alleles of *gcna-1* were generated using a co-CRISPR strategy as previously described (*Kim et al., 2014*), and outcrossed to N2. The nature of the alleles (on LGIII) is as follows: *ne4334*: 6007556/6007557–6007558/6007559 (2-bp deletion, stop codon after seven amino acids); *ne4356*: 6007278/6007279–6009026/6009027 (1748-bp deletion, removes ATG). The *ne4334* allele was amplified with primers zk328.4_F1 (5'-TTAGGACACCCTTGCTCTCG-3') and zk328.4_R1 (5'-GACGGTTCTGGTGTTTTGCT-3') and sequenced with zk328.4_F1. The *ne4356* allele was amplified with zk328.4_F3 (5'-CCGCCAATCAAAAATTTCAA-3') and zk328.4_R1, which generates a 2125-bp wildtype and 377-bp mutant band. Additionally, zk328.4_F1 and zk328.4_R1 are used to confirm the absence of a 762-bp wildtype product.

## C. *elegans* brood counts and male frequencies

Brood and male frequency counts were performed at 20 and 25°C. Briefly, animals were single picked at mid-L4 stage and followed with daily transfers until they produced no more progeny. Animals were counted and males were scored when the population on a progeny plate reached adulthood.

## Mice

All mouse studies were performed using a protocol approved by the Committee on Animal Care at the Massachusetts Institute of Technology (Protocol number: 0714-074-17).

## Mouse fertility testing

Six to eight week old male *Gcna*^DeltaEx4 /Y mice on a C57BL/6N background were housed with a single, wildtype C57BL/6N female of the same age for three months, and cages were monitored for births. Control animals were wildtype C57BL/6N males. The number of litters born and number of pups per litter were documented.

## Mouse epididymal histology

Mouse epididymal ducts were fixed overnight in Bouin's fixative at 4°C, then transferred to 70% ethanol before processing and embedding in paraffin. Five-micron sections were stained with hematoxylin and eosin before histological examination.

## Charge-hydropathy plot

N-terminal IDRs of GCNA orthologs were analyzed as in (*Uversky et al., 2000*) to determine their position in charge-hydropathy space relative to known disordered and ordered proteins. Briefly, hydrophobicity was calculated by the Kyte and Doolittle approximation using a window size of 5 amino acids. The hydrophobicity of individual residues was normalized to a scale of 0 to 1 in these calculations. The mean hydrophobicity is defined as the sum of the normalized hydrophobicities of all residues divided by the number of residues in the polypeptide. The mean net charge is defined as the net charge at pH 7.0, divided by the total number of residues.

## Amino acid composition analysis

Amino acid compositional analysis was carried out using Composition Profiler (*Vacic et al., 2007*) (RRID:SCR_014630, http://www.cprofiler.org) using the PDB Select 25 and the DisProt datasets as reference for ordered and disordered proteins, respectively. Enrichment or depletion in each amino acid type was expressed as $(C_x - C_{order})/C_{order}$, i.e., the normalized excess of a given residue's content in a query dataset ($C_x$) relative to the corresponding value in the dataset of ordered proteins ($C_{order}$). $C_x$ is either the GCNA ortholog dataset or the DisProt dataset, while $C_{order}$ is the PDB Select 25 dataset. Error bars represent fractional differences of the standard deviations of observed relative frequencies of bootstrapped samples of the datasets.

## Protein structure prediction and comparison

Protein structures of human GCNA, human Spartan, and *S. cerevisiae* Wss1 protease domains were predicted using I-TASSER (*Yang et al., 2015*) (RRID:SCR_014627, http://zhanglab.ccmb.med.umich.edu/), along with that of pdb4jiu (a minigluzincin protease whose structure has been determined [*López-Pelegrín et al., 2013*]). Model quality was evaluated using Swiss-model QMEAN model quality assessment (http://swissmodel.expasy.org/qmean, RRID:SCR_013032) (*Benkert et al., 2011*). The QMEANscore/Z-scores of the models were as follows: GCNA (0.651/−0.84), Spartan (0.665/−0.69), Wss1 (0.551/−1.56), and 4jiu (0.715/−0.24), where good quality models have a mean Z-score of -0.65. Using SaliLab Model Evaluation (https://modbase.compbio.ucsf.edu/evaluation/, RRID:SCR_004642) (*Melo et al., 2002*), the GCNA, Spartan, Wss1, and pdb4jiu models had GA341 reliability scores of 1.00/1.00/0.925/1.00 respectively, where a model with a score of greater than 0.7 is considered reliable and to have a probability of a correct fold of greater than 95%. We used FATCAT (*Ye and Godzik, 2004*) (RRID:SCR_014631, http://fatcat.burnham.org/) for pairwise protein structure comparison and obtained root-mean-square deviations (RMSDs) between 1.06 and 1.65 Angstroms as the average distance between the carbon atoms of the superimposed proteins. All structures were found to be significantly similar. Pairwise RMSDs and p-values not listed in the main figure are as follows: GCNA:Spartan (RMSD 1.65/p=7.81e-13), GCNA:Wss1 (RMSD 1.55/p=3.19e-9), Spartan:Wss1 (RMSD 1.41/p=4.05e-10). PDB files of structure models are available on request.

## Dendrogram of metalloprotease families

Multiple sequence alignments (in. a3m format) of the protease domains of GCNA, Spartan, and Wss1 families were created using the HHpred web server (*Söding et al., 2005*) at http://hhpred.tuebingen.mpg.de (RRID:SCR_010276) by providing a seed alignment that was subsequently expanded using HHblits with three iterations and an E-value inclusion threshold of 1E-30. The command line version of HHsearch was used to build HMMs using the hhbuild function. HHsearch was used to search a database of 14,837 PfamA curated protein families (RRID:SCR_004726) by comparing HMMs to HMMs using alignment and secondary structure scoring. With one exception (Pfam09768), the three HMMs significantly matched only metalloproteases within Peptidase Clan MA (as defined by the MEROPS database of peptidases [https://merops.sanger.ac.uk, RRID:SCR_007777]). Based on the probability score generated by HHsearch, a dissimilarity matrix of the proteins in the Clan MA was constructed, where the (*i*, *j*)-th entry gave the negative log probability that proteins *i* and *j* are

related. Probabilities of 0 were replaced with 0.001 to avoid infinity values when taking the logarithm. The proteins were heirarchically clustered by average linkage on the basis of this dissimilarity matrix using the 'linkage' and 'dendrogram' functions from the Python package SciPy (http://scipy.org, RRID:SCR_008058).

## SUMO interacting motif discovery

SUMO interacting motif discovery was carried out using the Eukaryotic Linear Motif resource (RRID: SCR_003085) (*Dinkel et al., 2016*) where a SUMO interacting motif is defined as [DEST]{0,5}.[VILPTM][VIL][DESTVILMA][VIL].{0,1}[DEST]{1,10}, representing a hydrophobic core surrounded by acidic or phosphorylatable residues, with the C terminal residues having a longer acidic stretch.

## Acknowledgements

We thank H Christensen, D deRooij, J Hughes, M Kojima, M Mikedis, P Nicholls, and K Romer for advice and comments on the manuscript; E Spooner at the Whitehead Proteomics Facility for mass spectrometry; the Koch Institute ES Cell and Transgenics Facility for assistance with C57BL/6N ES cell targeting; the WM Keck Biological Imaging Facility at the Whitehead Institute for use of the confocal microscope; G Welstead and M Goodheart for blastocyst injections; T DeCesare and T Endo for illustrations; A Godfrey for assistance with graphical representation using SciPy; P Tomancak, E Lecuyer, and H Krause for permission to reproduce fly embryo data; M Colaiacovo for generous instruction on *C. elegans* handling and immunohistochemistry; and M Martinez for technical assistance with targeting vector construction. Supported by the Howard Hughes Medical Institute and the Life Sciences Research Foundation.

## Additional information

### Funding

| Funder | Author |
|---|---|
| Howard Hughes Medical Institute | Michelle A Carmell<br>Gregoriy A Dokshin<br>Helen Skaletsky<br>Yueh-Chiang Hu<br>Kyomi J Igarashi<br>Daniel W Bellott<br>Peter W Reddien<br>Craig C Mello |
| Life Sciences Research Foundation | Michelle A Carmell |

The funders had no role in study design, data collection and interpretation, or the decision to submit the work for publication.

### Author contributions

MAC, GAD, Conception and design, Acquisition of data, Analysis and interpretation of data, Drafting or revising the article; HS, JCvW, VNU, Acquisition of data, Analysis and interpretation of data; Y-CH, Acquisition of data, Analysis and interpretation of data, Drafting or revising the article; KJI, Acquisition of data, Drafting or revising the article; DWB, Conception and design, Analysis and interpretation of data, Drafting or revising the article; MN, Conception and design, Contributed unpublished essential data or reagents; PWR, CCM, Analysis and interpretation of data, Drafting or revising the article; GCE, Acquisition of data, Contributed unpublished essential data or reagents; DCP, Conception and design, Drafting or revising the article

### Author ORCIDs

Peter W Reddien, http://orcid.org/0000-0002-5569-333X

David C Page, http://orcid.org/0000-0001-9920-3411

### Ethics

Animal experimentation: All mouse studies were performed using a protocol approved by the Committee on Animal Care at the Massachusetts Institute of Technology (Protocol number: 0714-074-17).

## Additional files

### Supplementary files

• Source code 1. Perl script for isoelectric point calculation.

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
