## [Decision Letter]

Thank you for submitting your article "A widely employed germ cell marker is an ancient disordered protein with reproductive functions in diverse eukaryotes" for consideration by *eLife*. Your article has been favorably evaluated by Marianne Bronner (Senior Editor) and two reviewers, one of whom, Yukiko M Yamashita, is a member of our Board of Reviewing Editors.

The reviewers have discussed the reviews with one another and the Reviewing Editor has drafted this decision to help you prepare a revised submission.

As you can see in the reviewers' comments, your manuscript was reviewed favorably and we would like to invite you to submit a revised manuscript. Because the reviewers are in agreement of supporting the publication of your manuscript and all comments are straightforward to address (essentially by editing the text), their comments are listed below without composing review summary. In preparing your revisions, please explain how you revised (how you incorporated or opted not to incorporate their suggestions).

We are looking forward to receiving the revised manuscript.

*Reviewer #1:*

In this manuscript, the authors reveal that two widely used antibodies against mouse germ cells, GCNA and Tra98, recognize the same ancient eukaryotic protein that they now name GCNA. They show that GCNA in mouse is a highly disordered protein consisting solely of intrinsically disordered regions (IDRs) thus precluding sequence/structure based homolog identification. They cleverly use synteny mapping to identify vertebrate orthologes of GCNA, which contain structured domains (protease, zinc finger and HMG box) in addition to the IDRs. They then utilize the structured domains in vertebrate GCNAs in order to identify GCNA orthologes in a sequence-dependent manner. The authors identify GCNA orthologes in all the major eukaryotic kingdoms and show that GCNA gene expression is specifically enriched in reproductive cells in a wide variety of eukaryotes. All of the above data (Figure 1–Figure 5) appears to be well supported and is clearly laid out within the manuscript.

Given the expression in reproductive cells across a wide swathe of eukaryotes, the authors next examine effects of GCNA mutation in mice and *C. elegans* (Figure 6–Figure 7), eukaryotes that diverged ~600 mya. In *C. elegans*, they generate two distinct GCNA mutants and show that these mutants exhibit a ~10-20% drop in brood size and a ~3-6 fold increase in X chromosome non-disjunction suggesting a defect in meiosis. In mice, the authors generate a GCNA mutant by removing GCNA exon 4 and show a striking loss of male fertility while female fertility is unaffected. These data are vital as they suggest an important role for GCNA in eukaryotic reproductive function. However, the discrepancy between a relatively small fertility defect in the *C. elegans* GCNA mutants in comparison to the almost complete sterility of male GCNA mutant mice is puzzling but not addressed. It is therefore unclear if GCNA is performing a similar role in both species and whether its function is conserved. To address this, a more comprehensive examination of mouse GCNA mutant phenotypes would be required: but such thorough characterizations are beyond the scope of this study, thus the text should be edited to convey this point (germ line function seems to be conserved, but the detail of their function in each species remains unclear).

Finally, in Figure 8, the authors suggest that GCNA is homologous to the DNA repair proteins, Wss1 and Spartan based on structural modeling, domain composition and the presence of SIM motifs in the unstructured domain. Based on predicted homology, the authors speculate in the Discussion that GCNA may bind sumoylated proteins and play a role in resolving meiotic double strand breaks. However, these speculations are not supported by experimental data and seem to add little to the manuscript. The manuscript does not even have data to show germ cells from GCNA mutants show phenotypes consistent with their idea, such as accumulation of DNA damages. Additionally, no evidence is provided to show that GCNA interacts with SUMOylated proteins. In the absence of such data, speculations on GCNA function seem to be without a solid foundation. Again, such in-depth characterizations of GCNA protein function are clearly beyond the scope of present study: thus again it is recommended that they edit the text (possibly removing figures) so as not to mislead the readers with the speculations that lack evidence.

*Reviewer #2:*

This is a very interesting paper describing the identification, functional analyses and evolutionary history of a protein family that is implicated in the earliest steps of meiosis and germ line specification in diverse eukaryotes. Interestingly, the proteins that they analyze were already well-known and longstanding antigenic markers for the germline, whose genes had not previously been revealed. The fact these genes are also widely distributed in diverse eukaryotes is the extra piece that makes this paper particularly compelling.

The data presented are solid and the overall conclusions are valid. I will not comment in detail on the experimental portions of the analyses except to say that I see no problems. However, I do find the bioinformatics to be mostly appropriate and their resulting inferences are overall supported. The paper will be of broad general interest to a diverse community including cell, molecular and evolutionary biologists--anyone interested in meiosis and the germ line. As such, the paper should be published in *eLife* pending some relatively minor revisions. Most of the comments below reflect suggestions to modify the precision of language, primarily as it reflects the evolutionary analyses and their inferences.

In general, I find some of the evolutionary language to be a bit loose for a scientific report. For example, in the abstract the authors cite "at least 600 million years", when nowhere in the paper do they cite a source for this number. Although this might be roughly correct, stating it in this way lends an unwarranted air of accuracy to their statements. Why not just say that it traces to the common ancestor of *C. elegans* and mammals? This issue is particularly problematic with Figure 4 in which the source(s) of information on divergence times are not specified and even if they were, should be not assumed to be very precise.

Specific comments:

In the fifth paragraph of the subsection “GCNA orthologs containing an IDR and a unique combination of conserved structured domains are present in every eukaryotic suprakingdom”, first sentence. I think that the authors mean "Superkingdom". They should also cite one of many possible papers on the definition of eukaryotic superkingdoms.

In the second paragraph of the subsection “GCNA is homologous to two protein families involved in replication-associated DNA repair”. Referring to Figure 8 as a phylogenetic tree is a bit misleading, as the method used doesn't really have the ability to recover the actual evolutionary relationships among these four protein families. I had to look in the Methods (subsection “Dendrogram of metalloprotease families') to reveal that the tree presented was prepared using a hierarchical clustering method. Yes, it's a dendrogram, but it has limited value in determining the evolutionary relationships. For example, there is little confidence in the relationships between the three eukaryotic protein groups and the prokaryotic one (e.g., what is the earliest branch and what are each others' closest relatives) making statements like "GCNA is in fact closer to Spartan" and "an exclusive cluster that diverged from each other in bacteria" lacking support. The dendrogram shown is a loose estimate at best, but since it’s not based on formal phylogenetic criteria should be discussed with caution.

Discussion, last paragraph. "GCNA is more ancient than all but *Piwi*, and predates the origin of a dedicated metazoan germline by a billion years." Again, this statement is overly strong and lacking in clear evidentiary support.

Figure 4. The tree shown is presumably the one referred to in the Methods (subsection “Phylogenetic tree construction”), but there is no information in the legend to indicate the data and methods used.

---

## [Author Response]

*Reviewer #1:*

[…]

*Given the expression in reproductive cells across a wide swathe of eukaryotes, the authors next examine effects of GCNA mutation in mice and C. elegans (Figure 6–Figure 7), eukaryotes that diverged ~600 mya. In C. elegans, they generate two distinct GCNA mutants and show that these mutants exhibit a ~10-20% drop in brood size and a ~3-6 fold increase in X chromosome non-disjunction suggesting a defect in meiosis. In mice, the authors generate a GCNA mutant by removing GCNA exon 4 and show a striking loss of male fertility while female fertility is unaffected. These data are vital as they suggest an important role for GCNA in eukaryotic reproductive function. However, the discrepancy between a relatively small fertility defect in the C. elegans GCNA mutants in comparison to the almost complete sterility of male GCNA mutant mice is puzzling but not addressed. It is therefore unclear if GCNA is performing a similar role in both species and whether its function is conserved. To address this, a more comprehensive examination of mouse GCNA mutant phenotypes would be required: but such thorough characterizations are beyond the scope of this study, thus the text should be edited to convey this point (germ line function seems to be conserved, but the detail of their function in each species remains unclear).*

We have modified the Discussion to address this concern.

*Finally, in Figure 8, the authors suggest that GCNA is homologous to the DNA repair proteins, Wss1 and Spartan based on structural modeling, domain composition and the presence of SIM motifs in the unstructured domain. Based on predicted homology, the authors speculate in the Discussion that GCNA may bind sumoylated proteins and play a role in resolving meiotic double strand breaks. However, these speculations are not supported by experimental data and seem to add little to the manuscript. The manuscript does not even have data to show germ cells from GCNA mutants show phenotypes consistent with their idea, such as accumulation of DNA damages. Additionally, no evidence is provided to show that GCNA interacts with SUMOylated proteins. In the absence of such data, speculations on GCNA function seem to be without a solid foundation. Again, such in-depth characterizations of GCNA protein function are clearly beyond the scope of present study: thus again it is recommended that they edit the text (possibly removing figures) so as not to mislead the readers with the speculations that lack evidence.*

As suggested by the Reviewer, we have abbreviated the discussion of possible biochemical roles of GCNA. That said, the discovery of sequence homology between GCNA, Wss1 and Spartan, as depicted in Figure 8, is a robust finding for which we provide computational methods and analytic results.

*Reviewer #2:*

[…]

*In general, I find some of the evolutionary language to be a bit loose for a scientific report. For example, in the abstract the authors cite "at least 600 million years", when nowhere in the paper do they cite a source for this number. Although this might be roughly correct, stating it in this way lends an unwarranted air of accuracy to their statements. Why not just say that it traces to the common ancestor of C. elegans and mammals? This issue is particularly problematic with Figure 4 in which the source(s) of information on divergence times are not specified and even if they were, should be not assumed to be very precise.*

To support our 600 million year statement, we have added three references (Discussion, first paragraph) that report the last common ancestor of *C. elegans* and mammals at approximately 560, 630, and 692 MYA. We have also specified, in the corresponding legend, the source of information on estimated divergence times used in Figure 4.

Specific comments:

*In the fifth paragraph of the subsection “GCNA orthologs containing an IDR and a unique combination of conserved structured domains are present in every eukaryotic suprakingdom”, first sentence. I think that the authors mean "Superkingdom". They should also cite one of many possible papers on the definition of eukaryotic superkingdoms.*

We have updated the language to reflect the more commonly used “Superkingdom,” and have added an appropriate citation.

*In the second paragraph of the subsection “GCNA is homologous to two protein families involved in replication-associated DNA repair”. Referring to Figure 8 as a phylogenetic tree is a bit misleading, as the method used doesn't really have the ability to recover the actual evolutionary relationships among these four protein families. I had to look in the Methods (subsection “Dendrogram of metalloprotease families”) to reveal that the tree presented was prepared using a hierarchical clustering method. Yes, it's a dendrogram, but it has limited value in determining the evolutionary relationships. For example, there is little confidence in the relationships between the three eukaryotic protein groups and the prokaryotic one (e.g., what is the earliest branch and what are each others' closest relatives) making statements like "GCNA is in fact closer to Spartan" and "an exclusive cluster that diverged from each other in bacteria" lacking support. The dendrogram shown is a loose estimate at best, but since it’s not based on formal phylogenetic criteria should be discussed with caution.*

In our original submission we did not make sufficiently clear the methods and formal phylogenetic criteria employed in constructing the tree shown in Figure 8. We have revised the legend to clarify this. The methods are described under the heading “Phylogenetic tree construction,” where we detail the use of PhyML to create a maximum likelihood tree based on formal phylogenetic criteria. We have also edited the language used to describe the relationships of the protein families.

*Discussion, last paragraph. "GCNA is more ancient than all but Piwi, and predates the origin of a dedicated metazoan germline by a billion years." Again, this statement is overly strong and lacking in clear evidentiary support.*

We have added references to support this statement.

*Figure 4. The tree shown is presumably the one referred to in the Methods (subsection “Phylogenetic tree construction”), but there is no information in the legend to indicate the data and methods used.*

We have updated the legend to indicate the source of the divergence time estimates and overall tree topology. We now state explicitly that divergence times are estimated.